# Iron Oxide Nanoparticles Combined with Static Magnetic Fields in Bone Remodeling

**DOI:** 10.3390/cells11203298

**Published:** 2022-10-20

**Authors:** Jiancheng Yang, Jiawen Wu, Zengfeng Guo, Gejing Zhang, Hao Zhang

**Affiliations:** 1Department of Spine Surgery, People’s Hospital of Longhua, Affiliated Hospital of Southern Medical University, Shenzhen 518109, China; 2Key Laboratory for Space Bioscience and Biotechnology, School of Life Sciences, Northwestern Polytechnical University, Xi’an 710072, China

**Keywords:** static magnetic fields, iron oxide nanoparticles, bone remodeling, osteoblast, osteoclast, bone regeneration, osteoporosis

## Abstract

Iron oxide nanoparticles (IONPs) are extensively used in bone-related studies as biomaterials due to their unique magnetic properties and good biocompatibility. Through endocytosis, IONPs enter the cell where they promote osteogenic differentiation and inhibit osteoclastogenesis. Static magnetic fields (SMFs) were also found to enhance osteoblast differentiation and hinder osteoclastic differentiation. Once IONPs are exposed to an SMF, they become rapidly magnetized. IONPs and SMFs work together to synergistically enhance the effectiveness of their individual effects on the differentiation and function of osteoblasts and osteoclasts. This article reviewed the individual and combined effects of different types of IONPs and different intensities of SMFs on bone remodeling. We also discussed the mechanism underlying the synergistic effects of IONPs and SMFs on bone remodeling.

## 1. Introduction

Due to in-depth research on magnetic nanomaterials in the biomedical field, medical magnetic nanomaterials now have special designs and standards. For clinical applications, safety is the most important factor, so not all magnetic nanomaterials have the potential to be clinically used in the future. Fe_3_O_4_ and γ-Fe_2_O_3_ solid-phase materials are easy to synthesize and have good chemical stability, magnetic properties, and biocompatibility [1]. Iron oxide nanoparticles (IONPs) have good biosafety and are the most promising magnetic nanomaterials in clinical practice [2]. Importantly, IONPs are the only inorganic functional nanomaterials that have been approved by the Food and Drug Administration (FDA) for clinical application. Over the years, many different IONPs have been evaluated in a wide variety of biomedical applications, including magnetic resonance imaging, tissue engineering, magnetic field drug targeting, and gene therapy [2]. IONPs also change some biological functions of cells. For example, studies have found that IONPs promote the polarization of tumor-associated macrophages into a pro-inflammatory type, thereby inhibiting tumor growth [3]. As a result, the medicinal value of IONPs should be developed based on the relevant biological effects rather than being used only as a nanomaterial.

Currently, the U.S. FDA has approved an IONP, named ferumoxytol (Feraheme), for the treatment of anemia in patients with chronic kidney disease; this is currently the only IONP used for clinical treatment [4]. Bone is a metabolically active tissue that is continuously being remodeled. Two major cell types involved in bone remodeling are osteoblasts, which are responsible for new bone tissue formation, and osteoclasts, which are responsible for bone resorption [5]. Over the past decade, a large number of in vitro studies demonstrated that IONPs promote osteoblast differentiation and inhibit osteoclast formation, whereas in vivo studies showed that IONPs accelerate bone defect repair and prevent bone loss [1]. Consequently, IONPs have certain application potential in bone tissue engineering and osteoporosis treatment.

As a noninvasive physical factor, clinical and animal studies showed that static magnetic fields (SMFs) have beneficial effects on osteoporosis, the nonunion of fractures, bone defect repair, and osteoarthritis. At the cellular level, SMFs promote the activity of osteoblasts and inhibit the differentiation of osteoclasts [6]. In recent years, an increasing number of studies have found that a combination of SMFs and IONPs is more effective than IONPs alone in promoting osteoblast differentiation and inhibiting osteoclast formation [7]. This review intended to summarize the effects of IONPs and/or SMFs on the activity of osteoblasts and osteoclasts. The mechanisms underlying the effect of IONPs combined with SMFs on bone remodeling are also discussed.

## 2. Effects of Iron Oxide Nanoparticles on Bone Remodeling

### 2.1. Effects of Iron Oxide Nanoparticles on Osteoblasts

As early as 2008, Pareta et al. [8] used IONPs to study osteoblast proliferation and confirmed that calcium-phosphate-coated γ-Fe_2_O_3_ nanoparticles significantly increased the density of osteoblasts (i.e., promoted cell proliferation). Subsequently, Tran et al. [9] showed that hydroxyapatite (HA)-coated Fe_3_O_4_ nanoparticles significantly promoted the production of alkaline phosphatase (ALP), collagen, and calcium in osteoblasts, indicating that IONPs promote osteogenic differentiation; the authors further investigated the mechanism by which IONPs promote osteoblast differentiation and found that IONPs adsorbed a large amount of fibronectin, which can increase the function of osteoblasts, and upregulated the expression of genes related to osteoblast differentiation [10]. In addition, the authors found that osteoblasts uptake HA-coated IONPs into the cytoplasm via receptor-mediated endocytosis and increase intracellular calcium levels, which may be another reason why HA-coated IONPs promote osteoblast functions [11]. However, other IONPs coated with noncalcium materials can also promote osteoblast activity. For example, Shi et al. [12] found that chitosan-coated IONPs promoted osteoblast proliferation, reduced cell membrane damage, increased ALP activity, and enhanced extracellular calcium deposition. Yin et al. [13] treated MG-63 cells, an osteoblast cell line, with Fe_3_O_4_ nanoparticles and found that cell proliferation and ALP activity were significantly promoted.

Stem cells have the ability to differentiate into a variety of cells, including osteoblasts. Xiao et al. [14] found that IONPs promoted cell proliferation, reduced apoptosis, increased ALP activity and mineralization nodule formation, and upregulated the expression of genes related to osteogenic differentiation in rat adipose-derived stem cells (ADSCs). Xia et al. [15] generated a scaffold by incorporating γFe_2_O_3_ and αFe_2_O_3_ nanoparticles into calcium phosphate cement (CPC). The authors found that human dental pulp stem cells (hDPSCs) seeded in this scaffold experienced increased osteogenic differentiation, ALP secretion, and mineral matrix synthesis compared with those seeded in scaffolds without IONPs, demonstrating that the osteogenic differentiation of hDPSCs was significantly promoted via the incorporation of IONPs into CPC. Similarly, Fe_3_O_4_-incorporated IONP–CPC scaffolds also enhanced the osteogenic differentiation of hDPSCs and promoted mandibular bone defect repair in rats [16]. In addition, studies have shown that IONPs have peroxidase activities [17]. Hydrogen peroxide (H_2_O_2_) was found to play an important role in the process of cell proliferation [18]. Huang et al. [19] treated human bone-derived mesenchymal stem cells (hBMSCs) with ferucarbotran (Resovist), an IONP approved for clinical liver MRI contrast agents, and found that ferucarbotran promoted cell proliferation by reducing intracellular H_2_O_2_ levels. These results indicate that IONPs have the ability to promote the proliferation and osteogenic differentiation of osteoblasts and stem cells in vitro.

Mechanistically, numerous studies have revealed that IONPs enhance osteogenic differentiation through multiple signaling pathways. Wnt signaling is a crucial pathway that mediates osteogenesis. In the classical Wnt pathway, β-catenin acts as a key transcriptional coactivator, transmitting extracellular signals to the nucleus to activate downstream target genes such as RUNX2 [20]. Xia et al. [21] revealed that γ-Fe_2_O_3_-loaded CPC scaffolds promoted the osteogenic differentiation of hDPSCs and significantly upregulated the gene expression of WNT1, RUNX2, ALP, COL1, and OCN. Moreover, β-catenin protein expression was increased, indicating that γ-Fe_2_O_3_-loaded CPC scaffolds activate Wnt/β-catenin signaling and downstream target genes. In osteoblast differentiation, increased osteoblastogenesis is dependent on the activation of β-catenin through the inhibition of GSK-3β [22], and the PI3K/Akt pathway can inhibit GSK-3β and activate β-catenin [23]. Yu et al. [24] developed a polysaccharide-based iron oxide nanoparticle (Fe_2_O_3_@PSC) and found that it has the ability to enhance osteoblast differentiation in MC3T3-E1 cells. A Western blotting assay showed that phosphorylated Akt, phosphorylated GSK-3β, and β-catenin were markedly upregulated. The authors proposed that Fe_2_O_3_@PSC promoted osteogenic differentiation by activating the Akt-GSK-3β-β-catenin signaling pathway. Mitogen-activated protein kinase (MAPK) includes three classic pathways, p38, extracellular signal-regulated kinase (ERK), and c-Jun N-terminal kinase (JNK), which play a key role in skeletal development and bone homeostasis, particularly in osteoblast commitment and differentiation [25,26]. Wang et al. [27] found that polyglucose-sorbitol-carboxymethyether (PSC) coated IONPs enhanced the expression of phosphorylated MEK1/2 and ERK1/2, indicating that IONPs activate the classic ERK-MAPK signaling pathway in hBMSCs. As a result, downstream genes of this pathway such as bone morphogenic protein (BMP2) and RUNX2 were upregulated to promote osteogenic differentiation. BMP2 is a signal molecule of the transforming growth factor-beta (TGF-β) superfamily and plays a crucial role in bone formation by activating the canonical Smad-dependent pathway or noncanonical-MAPK signaling pathway [28]. Lu et al. [29] fabricated a magnetic SrFe_12_O_19_ nanoparticle-modified mesoporous bioglass (BG) and chitosan (CS) porous scaffold (MBCS) to treat hBMSCs and found that this scaffold upregulated BMP2 and phosphorylated Smad1/5 expression and promoted the expression of osteogenic-related genes including RUNX2, OCN, COL1, and ALP, suggesting that magnetic MBCS scaffolds enhance osteogenic gene expression by activating the BMP-2/Smad signaling pathway. These in vitro studies indicate that IONPs or IONP-loaded scaffolds accelerate osteogenic differentiation through the Wnt/β-catenin, Akt-GSK-3β-β-catenin, MAPK, and BMP-2/Smad signaling pathways (Figure 1a).

In line with the in vitro studies, some in vivo studies have shown that scaffolds complexed with IONPs can promote bone formation. Hu et al. [30] implanted superparamagnetic IONP-loaded gelatinous sponges in rat incisor sockets, while gelatinous sponges without IONPs served as controls. Based on micro-CT and histological observations, the authors found greater formation of new bone compared with the blank control group at 4 weeks, suggesting that these IONPs induce active osteogenesis in vivo. Liao et al. [31] showed that PSC-coated IONPs promoted the differentiation of human precartilaginous stem cells (hPCSCs) into osteoblasts in vitro. In vivo, the authors incorporated IONP-labeled PCSCs in a novel methacrylated alginate and 4-arm poly(ethylene glycol)-acrylate (4A-PEGAcr) based interpenetrating polymeric printable network (IPN) hydrogel and implanted them into femoral defects in rats. The results of the micro-CT and histological analysis revealed that the implantation of IONP-labeled PCSCs significantly enhanced bone formation. Singh et al. [32] designed magnetic nanofibrous scaffolds by incorporating magnetic nanoparticles (MNPs) into poly(caprolactone) (PCL). The PCL–MNP nanofibrous scaffolds were subcutaneously implanted at the site of radial segmental defects. Histological images showed the favorable tissue compatibility and bone regenerative ability of the PCL–MNP nanofibers. Panseri et al. [33] obtained magnetic hydroxyapatite–collagen scaffolds via the nucleation of biomimetic hydroxyapatite and superparamagnetic IONPs on self-assembling collagen fibers. These magnetic scaffolds were implanted in rabbit tibial mid-diaphysis and distal femoral epiphysis. Histopathological screening showed that no inflammatory reaction occurred and that the bone-healing rate was significantly enhanced. Shuai et al. [34] constructed magnetic poly-l-lactide–polyglycolic acid (PLLA–PGA) scaffolds by incorporating Fe_3_O_4_ nanoparticles. The magnetic scaffolds were implanted into rabbit radius bone defects, and the results indicated that these scaffolds markedly induced substantial blood vessel tissue and new bone tissue formation at 2 months post-implantation, indicating that PLLA–PGA magnetic scaffolds offered excellent bone regeneration capabilities. Implantation of SrFe_12_O_19_–MBCS scaffolds into rat calvarial defects showed a significant increase in BMD and new bone areas at 12 weeks, suggesting that magnetic MBCS scaffolds enhance new bone regeneration in vivo [29]. Zhao et al. [35] incorporated nano-hydroxyapatite (nHAP) and Fe_3_O_4_ nanoparticles into the chitosan–collagen (CS–Col) organic matrix to obtain a magnetic CS–Col–Fe_3_O_4_–nHAP scaffold. A skull defect model of rats demonstrated that the CS–Col–Fe_3_O_4_–nHAP scaffold had better tissue compatibility and higher bone regeneration abilities when implanted into the skull defects compared with the control group. Overall, these magnetic scaffolds formed by incorporating IONPs seem to be promising for bone defect repair in the regenerative medicine field.

The above findings all indicate that IONPs promote osteoblast differentiation in vitro and accelerate bone formation and bone defect repair in vivo. However, not all IONPs are able to promote osteogenic differentiation. For example, citric-acid-coated IONPs reduced the cell viability of ADSCs and BMSCs and inhibited their adipogenic and osteogenic differentiation abilities [36,37], which may be related to the fact that citric acid can inhibit the proliferation of osteoblasts [38].

### 2.2. Effects of Iron Oxide Nanoparticles on Osteoclasts

Osteoclasts are differentiated from bone marrow macrophages (BMMs) under the induction of the receptor activator for nuclear factor-κ B ligand (RANKL) and macrophage colony-stimulating factor (M-CSF) and are the main cells that perform bone resorption. Compared with osteoblasts, there are fewer studies on IONPs on osteoclasts. Li et al. [39] treated mouse BMMs with PSC-coated IONPs and HA-coated IONPs, finding that both IONPs significantly inhibited osteoclast formation and downregulated osteoclast-differentiation-related gene expression. Postmenopausal osteoporosis is a disease characterized by reduced BMD, damaged bone microstructure, and increased bone fragility induced by increased osteoclast activity [40]. Bilateral ovariectomy (OVX) in animals is the most commonly used model used to mimic postmenopausal osteoporosis. Liu et al. [41] found that ferucarbotran and Feraheme inhibited the differentiation of mouse BMMs into osteoclasts, whereas the intravenous injection of two types of IONPs markedly inhibited bone resorption and OVX-induced bone loss in OVX mice. Zheng et al. [42] also revealed that PSC-coated IONPs inhibited osteoclast differentiation and prevented bone loss caused by OVX. In addition, the authors prepared IONPs loaded with alendronate, a drug used for the treatment of osteoporosis, and found that IONPs could target the bone tissue; the IONPs’ ability to inhibit bone loss was significantly better than that of alendronate alone. Iron is an essential element involved in multiple life activities of the human body, including bone metabolism [43]. However, excessive iron can induce osteoporosis by activating osteoclast activity [44]. Yu et al. [24] showed that the PSC-loaded Fe_2_O_3_ nanoparticles inhibited osteoclast differentiation of Raw 264.7 cells in vitro and prevented iron-accumulation-related osteoporosis in vivo.

During osteoclast differentiation, RANKL binds to its receptor RANK on BMMs and activates many signaling pathways, including MAPKs (ERK, JNK, and p38) and nuclear factor-kB (NF-kB), by recruiting the signaling-adaptor molecule TNF receptor-associated factor 6 (TRAF6) [45]. Among them, the ubiquitination of TRAF6, which involves the important adaptor protein p62 and deubiquitinase cylindromatosis (CYLD), is a key process [46]. Liu et al. [41] revealed that IONPs enhanced the expression of p62, which resulted in the recruitment of CYLD and promoted the deubiquitination of TRAF6. Moreover, the downstream MAPK and NF-κB signaling pathway was inhibited, leading to decreased expression of osteoclastogenesis-related genes, including NFATC1, ACP5, CALCR, CTSK, and c-SRC. Similarly, Yu et al. [24] demonstrated that Fe_2_O_3_@PSC nanoparticles suppressed osteoclast differentiation by inhibiting the MAPK and NF-κB pathways in vitro. Therefore, IONPs can inhibit osteoclast differentiation through the retardation of MAPK and NF-κB signaling pathways (Figure 1b).

## 3. Effects of Static Magnetic Fields on Bone Remodeling

An SMF is a type of magnetic field with a constant magnetic field strength and direction. SMFs have been subjected to many years of fundamental and clinical research history in the field of bone biology.

### 3.1. Effects of Static Magnetic Fields on Osteoblasts

According to the strength of the magnetic field, SMF can be classified as a hypomagnetic field (HyMF, <5 μT, commonly found in outer space), weak SMF (5 μT–1 mT, such as a geomagnetic field), moderate SMF (1 mT–1 T, such as common permanent magnets), and high SMF (>1 T, such as MRI and superconducting magnet) [47]. Numerous studies have shown that SMFs with different magnetic field strengths have different effects on osteoblast proliferation and differentiation [48].

To date, only one study has reported the effects of HyMF on osteoblasts. Yang et al. [49] exposed MC3T3-E1 cells to HyMF of 500 nT (generated by a magnetic shielding box made of a permeability alloy) for 2 and 8 days. The results showed that ALP activity, mineralization nodule area, and osteogenic gene expression were markedly decreased compared with the geomagnetic field, indicating that HyMF attenuated osteogenic differentiation.

Moderate SMFs are the easiest to obtain and the most common in daily life. Thus, most of the magnetic field strengths used in studies exploring the effects of SMFs on osteogenic differentiation focused on a moderate intensity range. An SMF of 15 mT promoted proliferation, ALP activity, and mineralized nodule formation in hBMSCs in a time-dependent manner and upregulated the expression of osteogenic marker genes, indicating that 15 mT SMF promotes osteogenic differentiation and biomineralization in hBMSCs [50]. Similarly, osteoblastogenesis was also enhanced by an SMF of 0.2, 0.4, and 0.6 T in BMSCs from rats or mice [51,52]. Zheng et al. [53] showed that an SMF of 1 mT increased cell proliferation, osteogenic differentiation, and biomineralization in hDPSCs. Yamamoto et al. [54] isolated rat calvaria cells and induced their osteoblastic differentiation under a 160 mT SMF. The results showed a significant increase in the total area and number of bone nodules, calcium content, and ALP activity, suggesting that an SMF of 160 mT accelerated the osteogenic differentiation of primary rat osteoblasts. In pre-osteoblast cell lines such as MG63 and MC3T3-E1 cells, studies have also shown that a moderate SMF stimulates osteoblastic differentiation and biomineralization [55,56]. Overall, these results suggest that moderate SMFs of different intensities can promote osteogenic differentiation in stem cells, primary osteoblasts, and pre-osteoblast cell lines.

Currently, the strongest human MRI commercially available in the world has reached 10.5 T [57], whereas MRIs for small animals can reach as high as 21.2 T [58]. Therefore, it is necessary to explore the effects of high SMFs on bone remodeling. As early as 2002, Kotani et al. [59] exposed MC3T3-E1 cells to an SMF of 8 T generated by using a superconducting magnet and found that high SMFs remarkably enhanced osteoblast differentiation, manifested as increased ALP activity and mineralization nodule formation. Recently, we exposed MC3T3-E1 cells to 2 and 16 T high SMFs generated by superconducting magnets and also found that both 2 and 16 T high SMFs significantly promoted osteoblastic differentiation and bone formation [49,60]. These results demonstrate that a high SMF has no toxic effect on osteoblasts and can promote osteoblastic differentiation.

Osteocytes, the most abundant cells in bone tissue, are descended from mature, matrix-producing osteoblasts and play a key role in regulating bone remodeling [61]. However, there are few studies on the effects of SMFs on osteocytes. Recently, we illustrated that 16 T SMF elevated cellular viability, decreased apoptosis, enhanced the fractal dimension of the cytoskeleton, and changed the secretion of cytokines. Additionally, an SMF-modulated cellular iron metabolism may be involved in altering the biological effects of osteocytes under 16 T SMF exposure [62].

### 3.2. Effects of Static Magnetic Fields on Osteoclasts

Contrary to the inhibition effects of HyMF on osteoblast differentiation, previous studies have shown that HyMF inhibits the differentiation of osteoclasts. Tartrate-resistant acid phosphatase (TRAP) is a marker of mature osteoclasts [63]. Zhang et al. [64] exposed pre-osteoclasts Raw264.7 to an HyMF of 500 nT for 2 and 4 days. The results showed that more TRAP-positive multinucleated cells, stronger TRAP activity, and higher expression of osteoclast marker genes in Raw264.7 cells were induced under HyMF compared with those under a geomagnetic field. Moreover, more bone resorption pores were induced after 10 days of differentiation induction by seeding Raw264.7 cells in Osteo Assay Surface 96-well plates exposed to HyMF. These data clearly show that HyMF stimulates osteoclast differentiation and its bone resorption capacity.

Conversely, an inhibitory effect on osteoclast differentiation was revealed under moderate and high SMFs. Kim et al. [65] demonstrated that an SMF of 15 mT inhibited the differentiation of mouse BMMs into osteoclasts and reduced TRAP activity and bone resorption capacity. Zhang et al. [64] found that a 16 T SMF exposure significantly inhibited the differentiation of Raw264.7 cells into osteoclasts and reduced the expression of genes related to osteoclast differentiation. We also showed that a 16 T SMF inhibited osteoclastic formation and bone resorption abilities, which may be related to the regulation of cellular iron metabolism by a high SMF [66].

## 4. Effects of Iron Oxide Nanoparticles Combined with Static Magnetic Fields on Bone Remodeling

In recent years, an increasing number of studies have attempted to combine IONPs and scaffolds incorporating IONPs with SMFs to investigate their effects on the activities of osteoblasts and osteoclasts. The effects of such combined applications are often significantly better than those of single IONPs or magnetic field effects (Table 1).

### 4.1. Effects of Iron Oxide Nanoparticles Combined with Static Magnetic Fields on Osteoblasts

Numerous studies have reported the effects of IONPs combined with SMFs of different strengths on osteoblast differentiation in vitro and bone regeneration in vivo. Huang et al. [79] deposited Fe_3_O_4_ nanoparticles and polydopamine (PDA) onto the surfaces of 3D-printed pTi scaffolds (Fe_3_O_4_/PDA@pTi) and found that the cell proliferation and osteogenic differentiation of hBMSCs was promoted in vitro, whereas the new bone formation of femoral bone defects in rabbits was enhanced under an SMF of 15 mT. He et al. [83] indicated that the addition of graphene oxide to Fe_3_O_4_ could promote the osteogenic abilities of rat BMSCs through the Wnt/β-catenin pathway under an SMF of 15 mT. Zhao et al. [82] encapsulated SPIONs into the PLGA microspheres to form three types of PLGA microspheres (PFe-I, PFe-II, and PFe-III). The authors found that the osteogenic differentiation of rat BMSCs was significantly promoted by PFe-II. Afterward, the authors implanted PFe-II microspheres into the defect zone of the rat femoral bone, followed by exposure to an external SMF of 50 mT. The results showed that the bone mineral density (BMD), trabecular thickness (Tb.Th), and bone tissue volume/total tissue volume (BV/TV) at the defect zone were significantly higher than those of the PFe-II microspheres alone. Xia et al. [16] revealed that a combination of IONP–CPC scaffolds with SMF effectively accelerated the cell proliferation and osteogenic differentiation of hDPSCs, leading to fourfold higher new bone regeneration compared with the CPC control in vivo. Marycz et al. [77] showed that exposure of a thermoplastic polyurethane (TPU) and PLA polymer doped with IONPs to an SMF resulted in improved osteogenic differentiation of mouse ADSCs. Yun et al. [73] demonstrated that the combined application of SMF and PCL-IONP scaffolds synergistically promoted osteoblastic ALP activity and the expression of osteogenesis-related genes in primary mouse calvaria osteoblasts, as well as accelerated bone formation at the bone defect sites in mice. Cai et al. [71] treated MC3T3-E1 cells with PLA-coated Fe_3_O_4_ nanoparticles and exposed them to an SMF of 100 mT, finding that osteoblastic adhesion, proliferation, ALP activity, and calcium content were markedly enhanced in the magnetically exposed group compared with those in the nonmagnetically exposed group. Zeng et al. [68] demonstrated that HA-IONP scaffolds significantly promoted the cell proliferation, ALP activity, and osteocalcin production of MC3T3-E1 cells in the presence of an external SMF. Tang et al. [80,81] demonstrated that magnetic CoFe_2_O_4_/P(VDF-TrFE) or ZnFe_2_O_4_ coatings with the assistance of a 200 mT SMF could promote the cell adhesion, early proliferation (3 days), and osteogenic differentiation of MC3T3-E1 cells. Overall, these results suggest that IONPs in combination with SMFs synergistically promote osteogenic differentiation in various stem cells, primary osteoblasts, and osteoblast cell lines and accelerate bone defect repair in rats and mice.

In addition, several studies have found that, while IONPs alone do not have a significant effect on osteoblast differentiation, osteogenesis is significantly promoted in the presence of an external SMF. For example, Jiang et al. [72] showed that BSA-coated Fe_3_O_4_ nanoparticles had no effect on ALP activity, calcium deposition, or protein expression related to osteogenic differentiation in the BMSCs of rats, whereas osteogenic differentiation was significantly promoted under the stimulation of a 1 T SMF. Zhuang et al. [75] coated type I collagen with IONPs at the bottom of the cell culture plate and seeded MC3T3-E1 cells on the plate, finding that the coatings had no significant effect on osteoblast differentiation. However, the ALP activity, mineralization deposition, and osteogenesis-related gene expression were markedly enhanced when exposed to an SMF of 100 mT. These data further demonstrate the superior enhancement of osteogenic differentiation when using SMFs in combination with IONPs.

### 4.2. Effects of Iron Oxide Nanoparticles Combined with Static Magnetic Fields on Osteoclasts

Currently, only Marycz K et al. have studied the effects of IONPs combined with SMFs on osteoclastogenesis. The authors fabricated α-Fe_2_O_3_/γ-Fe_2_O_3_ and investigated its effects alone and in combination with an SMF of 200 mT on differentiated MC3T3-E1 cells and Raw264.7 cells [7]. The results showed that α-Fe_2_O_3_/γ-Fe_2_O_3_ promoted osteoblast differentiation and inhibited osteoclast activity; these effects were enhanced when an SMF was applied to the cell culture environment. Moreover, α-Fe_2_O_3_/γ-Fe_2_O_3_ increased the expression of BAX, p21, and Casp-9 and reduced the mitochondrial membrane potential in differentiated Raw264.7 cells, indicating that the mitochondrial apoptosis pathway is activated in osteoclasts. Subsequently, the authors fabricated a novel Co_0_._5_Mn_0_._5_Fe_2_O_4_@PMMA nanoparticle and investigated its potential utility in the treatment of osteoporosis using pre-osteoblasts MC3T3-E1 and pre-osteoclasts 4B12 in the presence of an SMF. The results showed that these nanoparticles promoted osteoblastic differentiation through the activation of the OPN–BGLAP2–DMP1 axis and modulated osteoclastogenesis [84]. Recently, the authors functionalized Ca_5_(PO_4_)_3_OH/Fe_3_O_4_ with miR-21/124 to treat MC3T3-E1 and pre-osteoclasts 4B12 cells under an SMF of 200 mT. The results showed that osteogenetic markers were activated, whereas osteoclast differentiation markers were reduced [85]. These results indicate that a combination of SMFs with IONPs may influence the proper regeneration of osteoporotic bone by restoring the balance between osteoblasts and osteoclasts. However, more studies are needed to determine the effects of SMFs combined with IONPs on osteoclastogenesis.

## 5. Mechanism of Static Magnetic Field Enhanced the Biological Effects of Iron Oxide Nanoparticles

### 5.1. Micromagnetic Field Effects

The most notable advantage of IONPs is that they are superparamagnetic; that is, IONPs have a strong magnetic response when exposed to an external magnetic field, and there is no remanence in the particles when the external magnetic field is removed [86]. IONPs are nonmagnetic on a macroscopic scale. However, IONPs exposed to external magnetic fields can be rapidly magnetized to saturation, at which point the IONPs behave somewhat like micromagnets. Then, the cells become exposed to the ensuing micromagnetic field. Sun et al. [70] speculated that the micromagnetic fields generated by assemblages of IONPs may promote the differentiation of primary mouse BMSCs into osteoblasts. Indeed, when IONPs, including the assemblages and natural aggregates, were subjected to 120 °C for over 8 h to demagnetize them, the assemblages of IONPs under SMFs of different magnetic field intensities exhibited almost the same effects as those of nonmagnetic-field exposed IONPs on the BMSCs [70]. These results partially validate the hypothesis that the synergistic effects of IONPs and SMFs on cells are due to micromagnetic field effects.

### 5.2. Mechanical Stimulation

Mechanical stimulation plays a key role in regulating bone remodeling [87]. Multiple mechanical forces stimulate the proliferation and differentiation of osteoblasts and BMSCs [88] while inhibiting osteoclastic differentiation [89,90]. It was hypothesized that local mechanical stress might be induced by the combined application of magnetic materials and an SMF, which could further improve cellular behavior [91]. Recently, the forces of magnetic nanoparticles in the magnetic field generated by permanent magnets were discussed in detail by Blümler P [92]. Due to magnetization, magnetic nanoparticles can be regarded as magnetic dipoles. In an inhomogeneous magnetic field (mainly referring to those generated by permanent magnets), magnetic nanoparticles are subject to the attractive force of the external magnetic field and the magnetic dipolar interactions between the magnetic nanoparticles. This magnetically actuated force causes deformation of the magnetic matrix containing IONPs in the SMF, which was directly characterized via in situ scanning using atomic force microscopy (AFM) by Hao et al [76]. Distorted magnetic matrixes and magnetic scaffolds can offer mechanical stimuli on cells under SMFs, which may underpin the synergistic effects of IONPs and SMFs to enhance osteogenic differentiation [70,76,93,94]. Moreover, when IONPs are associated with cells, compressive and tensile forces are induced on the cell membrane under the action of an external magnetic field, resulting in a series of cellular biochemical reactions including changes in intracellular calcium levels [95].

Mechanotransduction converts physical forces acting on cells into internal biochemical signals through multiple mechanosensing pathways. Myosin-II and Rho-associated protein kinase (ROCK) are two important mechanosensitive proteins that are essential for regulating the normal differentiation of osteoblasts [96,97]. Jiang et al. [72] treated BMSCs with nonmuscle myosin-II inhibitor blebbistatin or ROCK inhibitor Y-27632 and exposed the cells to BSA-doped Fe_3_O_4_ particles together with an SMF. The authors found that the osteogenic differentiation of BMSCs stimulated by the combined effects of IONPs and the magnetic field was almost completely inhibited. These results again suggested the importance of mechanical stimulation in the osteogenic differentiation of IONPs combined with SMFs.

Integrins, the predominant molecular transducers of force [98], can stimulate osteogenic differentiation potential in MSCs when acted on by magnetic forces [99]. Kasten et al. [100] demonstrated that the integrin-mediated mechanical forces caused by a magnetic field promote the expression of COL1, which is involved in the osteogenesis in MSCs. The obtained results showed that the mechanotransduction process of IONPs and SMFs might be correlated with integrin overexpression. Marycz et al. [7] found a substantial increase in the expression of integrin alpha 3 (INTa3) in MC3T3-E1 cells after the stimulation of α-Fe_2_O_3_/γ-Fe_2_O_3_ nanocomposites and SMFs. The authors also demonstrated that TPU/PLA-doped IONPs and an SMF significantly upregulated the expression of integrin alpha 2 (INTa2), INTa3, and integrin beta 1 (INTβ1) in ADSCs [77]. Similarly, the expression of INTa2 and INTβ1 in IONP-treated MC3T3-E1 cells was greatly upregulated with the assistance of SMFs [80]. Moreover, SMFs greatly increased the expression level of the phosphorylation of ERK (p-ERK) molecules in MAPK signaling pathways, indicating that integrin-mediated MAPK pathways represent the mechanism underlying increased osteogenic differentiation under the condition of magnetic nanocomposite coatings combined with SMFs [80]. Yun et al. [73] revealed that integrin downstream signaling molecules, including p-FAK, p-paxillin, phosphorylation of protein kinase 38 (p38), ERK1/2, c-Jun-N terminal kinases (JNKs), and Roh A, were markedly activated in primary mouse calvarial osteoblasts under the condition of IONP-incorporated magnetic scaffolds combined with an SMF. In addition, BMP2 was promoted by the magnetic force stimuli and its downstream signaling; the phosphorylation of Smad1/5/8 is also activated [73]. There is extensive cross-talk between integrins and TGFβ, i.e., RGD-binding integrins can activate latent TGFβ [101]. The TGFβ and BMP signaling pathways regulate the expression of osteogenic genes (e.g., RUNX2) by sharing common canonical Smad-dependent pathways and noncanonical Smad-independent signaling pathways (e.g., MAPK) [102]. Huang et al. [79] showed that SMFs enhanced osteogenic differentiation in vitro and new bone formation in vivo through a synergistic effect with Fe_3_O_4_/PDA@pTi. Moreover, the results of the Western blotting analysis verified that the overexpression of TGF-βRI, TGF-βRII, phosphorylated Smad2/3, and Smad4 was more significant in the Fe_3_O_4_/PDA@pTi with the SMF group than in the pTi group. Therefore, the activated integrin-MAPKs, BMP2-Smads, and TGFβ-Smads signaling pathways are potential molecular mechanisms for the enhancement of osteogenesis via the magnetic forces of IONPs and SMFs.

Piezo-type mechanosensitive ion channel component 1 (Piezo1), a recently identified bona fide mechanotransducer, confers mechanosensitivity on osteoblasts and plays a critical role in bone formation [103]. To date, the only method known to activate Piezo1 is mechanical stimulation, except for the chemical agonist Yoda1 [104]. Recently, Hao et al. [76] revealed that the expression of Piezo1 was significantly upregulated upon exposure to an SMF in a dose-dependent manner in MC3T3-E1 cells cultured on magnetic nanocomposites. In contrast, in the absence of SMF exposure, only small changes in the expression levels of Piezo1 were found in cells. The gene expression of BMP2 was also increased, indicating that BMP2 signaling is a potential target of Piezo1 in osteogenic differentiation. Indeed, it was demonstrated that after mechanical loading, Piezo1 expression is upregulated, subsequently promoting BMP2 expression and osteoblast differentiation [105]. These results further suggest that IONP-treated cells were subjected to mechanical stress in the presence of an SMF.

### 5.3. Increases in Intracellular Iron Oxide Nanoparticles

An external magnetic field can align the IONPs, and, once the magnetic field is removed, the IONP can be randomized. However, in colloidal suspensions of IONPs, applying a magnetic field may cause the IONPs to agglomerate [106]. Alterations in the physicochemical properties of IONPs strongly affect their biological properties. Thus, the agglomeration of IONPs due to magnetic fields can change their well-recognized biological effects [107]. Indeed, significant changes in the sizes of IONPs (from less than 100 to 300 nm) and zeta potential induced by the aggregation of particles can cause changes in the uptake of IONPs by cells [107]. Moreover, the release of IONPs from cells is inhibited by an SMF, resulting in nearly twice the number of intracellular IONPs as those without an SMF [72]. The end result is that the greater the uptake of intracellular IONPs under the stimulation of an SMF, the greater the osteogenic differentiation of BMSCs [72]. These findings are consistent with those of another study using IONP–CPC scaffolds in cultures with hDPSCs in the absence or presence of an SMF [16]. The IONP–CPC scaffolds combined with the magnetic field resulted in a significant increase in the iron content inside the cells and the osteogenic differentiation of hDPSCs [16]. Therefore, the changes in the physicochemical properties and cellular endocytosis of IONPs under the application of magnetic fields can significantly enhance cellular behavior and bone regeneration capacity.

Ultimately, existing studies on the mechanisms underlying the effects of SMFs combined with IONPs on bone cells mainly focused on osteoblasts or stem cells. In Figure 2, we summarize the potential mechanisms by which combinations of SMFs and IONPs impact osteogenic differentiation.

## 6. Conclusions and Prospects

This review discussed the effects of different types of IONPs and different intensities of SMFs on bone remodeling. When exposed to IONPs, pre-osteoblasts and stem cells experience a marked improvement in cell proliferation and osteogenic differentiation, whereas pre-osteoclasts and BMMs display a significant obstruction in osteoclastic differentiation. This result is further supported by in vivo findings, showing that osteoblasts and osteoclasts internalize the IONPs, yielding superior bone regeneration and weaker bone resorption. Moreover, the use of an external SMF synergistically enhanced osteogenic differentiation and inhibited osteoclastic differentiation, which potentially attributed to micromagnetic field effects, magnetically actuated mechanical stimuli, and increased intracellular IONP levels in the presence of an SMF. The combined application of IONPs and an SMF can be a noninvasive and convenient form of therapy to promote bone regeneration. However, there remain some interesting unresolved questions regarding the effects of IONPs and SMFs on bone remodeling that deserve exploration in the future.

Osteoporosis is a degenerative bone disease commonly related to aging. The prevalence of osteoporosis is steadily growing due to demographic changes toward an increasingly aged population. Osteoclast-mediated increases in bone resorption are the main cause of osteoporosis [108]. Although the effects of SMFs and IONPs alone on osteoclasts have been partially reported, there are few studies on the combined effects of IONPs and SMFs. Therefore, further research is needed to explore the effects of a combination of SMFs and IONPs on osteoclast differentiation in vitro and osteoporosis in vivo.

Osteocytes descend from osteoblasts encapsulated by a mineralized bone matrix and constitute over 90% of bone cells in the adult skeleton [109]. These cells act as a coordinator in bone remodeling, modulating the differentiation and function of osteoclasts and osteoblasts through distinct signaling pathways, including the RANKL–OPG and SOST–Dkk1–Wnt axes [110]. Although osteocytes are very important for bone remodeling, there is no report on the effects of IONPs on osteocytes. Osteocytes are mechanosensitive cells that sense mechanical stimuli through their lacunar–canalicular system (LCS), which extends throughout the bone matrix, and then transmit signals to osteoblasts, osteoclasts, and other osteocytes [111,112]. Therefore, we speculate that mechanical stimulation from IONPs combined with SMFs can change the secretion of signaling molecules in osteocytes and indirectly regulate the differentiation and function of osteoblasts and osteoclasts. This subject would be a rewarding direction for future studies, as osteocytes play a crucial role in bone homeostasis.

Previous studies on IONPs related to bone repair or osteoporosis have mainly focused on the biological effects of IONPs in animal or cell experiments. Thus, rigorous clinical trials in humans are needed before translating these findings into clinical practice. Moreover, toxicity is the most important evaluation index in clinical therapy. However, existing studies fail to evaluate the safety of IONPs in vivo. The toxicity of IONPs should be considered in a dose-, treatment-, and time-dependent manner [113]. Therefore, the absorption, distribution, metabolism, and toxicity of IONPs after implanting a composite scaffold containing IONPs in vivo should be explored in future studies.

## Figures and Tables

**Figure 1 cells-11-03298-f001:**
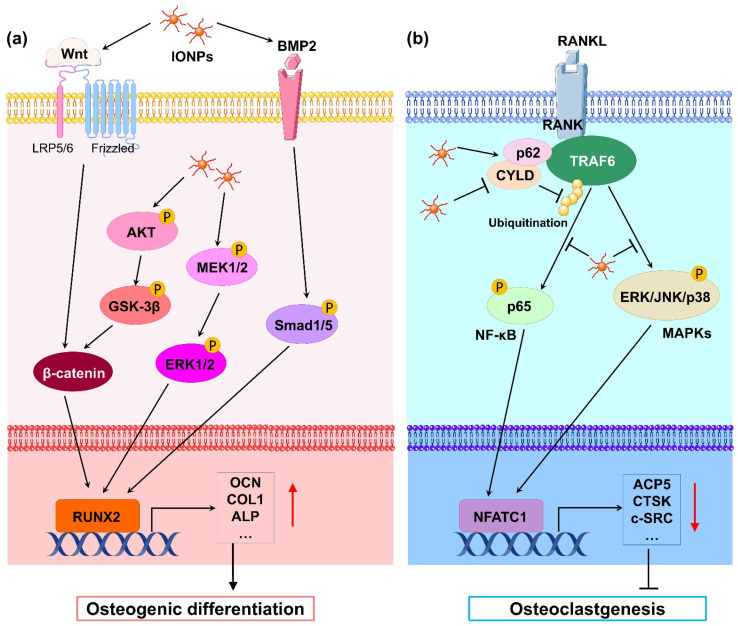
(**a**) Schematic illustration of IONP-promoted osteogenic differentiation in osteoblasts and stem cells. Classical Wnt/β-catenin, Akt-GSK-3β-β-catenin, MAPK, and BMP-2/Smad signaling pathways are activated by IONPs. Thus, osteoblastogenesis-related gene transcription downstream is markedly promoted, leading to enhancement of osteogenic differentiation. (**b**) Schematic illustration of IONP-attenuated osteoclastogenesis in BMMs. IONPs upregulated p62 expression by increasing the binding of CYLD to the TRAF6–p62–CYLD complex, resulting in repressive ubiquitination of TRAF6 and inhibition of RANKL-induced signaling pathway such as NF-κB and MAPK signals. As a result, transcription of osteoclastogenesis-related genes was obviously blocked, leading to blockage of osteoclastogenesis.

**Figure 2 cells-11-03298-f002:**
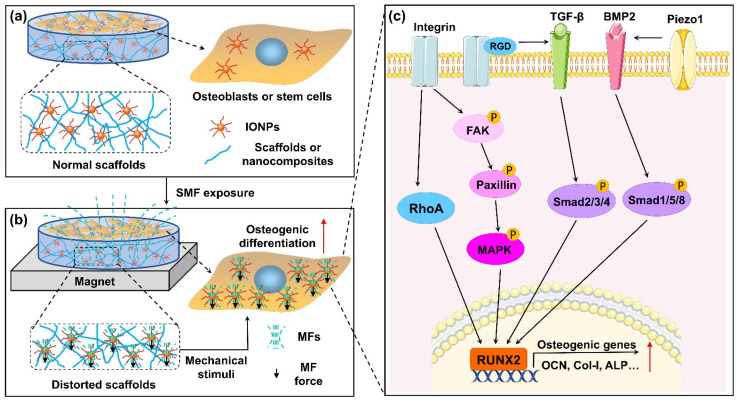
Schematic depiction of combination of magnetic field (MF) and IONP-enhanced osteogenic differentiation. (**a**) Osteoblasts or stem cells co-cultured with IONPs or IONP-doped scaffolds without SMF. (**b**) Osteoblasts or stem cells co-cultured with IONPs or IONP-doped scaffolds under SMF. IONPs exposed to SMF can be magnetized and generate magnetic force. On one hand, magnetic force induces deformation of scaffolds, which generates mechanical stimulation on cells cultured on scaffolds. On the other hand, magnetic forces from intracellular IONPs may directly cause deformation of the cell membrane. In addition, cells take in more IONPs, and magnetized IONPs generate micro-MF effects on the cells. (**c**) Combination of MF and IONPs activated the integrin-RhoA, integrin-MAPK, and TGFβ/BMP2 Smad signaling pathways. Subsequently, transcription of osteoblastogenesis-related genes was markedly enhanced. Furthermore, RGD-binding integrins are able to activate latent TGF-β, and activated Piezo1 can upregulate BMP2 expression.

**Table 1 cells-11-03298-t001:** Effects of iron oxide nanoparticles combined with static magnetic fields on bone remodeling.

Cell Type	Animal Model	Magnetic Field Intensity	IONPs	Outcome	Ref.
Pre-osteoblasts MC3T3-E1	No application	0.9–1.0 mT	HA- and PLA-coated γ-Fe_2_O_3_	Promoting cell proliferation and ALP secretion	[67]
Pre-osteoblasts MC3T3-E1	No application	Not clear	HA doped with Fe_3_O_4_	Enhancing osteoblast proliferation, ALP activity, and osteocalcin synthesis	[68]
No application	Rabbit model of lumbar transverse defects.	0.05–25 mT	HA and PLA doped with γ-Fe_2_O_3_	Accelerating new bone tissue formation	[69]
Mouse BMSCs	No application	20–120 mT	Bare γ-Fe_2_O_3_	Enhancing osteogenic differentiation	[70]
Pre-osteoblasts MC3T3-E1	No application	100 mT	PLA doped with Fe_3_O_4_	Promoting the proliferation and osteogenic differentiation	[71]
Rat BMSCs	No application	1 T	BSA doped with Fe_3_O_4_	Elevating ALP activity, calcium deposition, and expressions of osteogenic markers	[72]
Primary mouse calvarium osteoblasts	Mouse model of calvarium defects	15 mT	PCL doped with Fe_3_O_4_	Enhancing osteoblastic differentiation in vitro and the new bone formation	[73]
Pre-osteoblasts MC3T3-E1	Beagle dog with femur transverse defect	200 mT	HA doped with Fe_3_O_4_	Increasing cell proliferation in vitro and bone healing in vivo	[74]
Pre-osteoblasts MC3T3-E1	No application	100 mT	Mineralized collagen doped with IONPs	Enhancing ALP activity, calcium deposition, and expressions of osteogenic genes	[75]
Pre-osteoblasts MC3T3-E1	No application	70–80 mT	Oleic acid and PLGA doped with IONPs	Promoting cell attachment and osteogenic differentiation	[76]
hDPSCs	Rat model of mandible defects	35 ± 5 mT	CPC doped with γ-Fe_2_O_3_	Enhancing osteogenic differentiation in vitro and bone formation in vivo	[16]
Pre-osteoblasts MC3T3-E1	No application	200 mT	α-Fe_2_O_3_/γ-Fe_2_O_3_ nanocomposite	Enhancing expression of crucial markers for osteogenesis	[7]
Mouse ADSCs	No application	200 mT	TPU and PLA doped with Fe_2_O_3_	Enhancing osteogenic differentiation of ADSCs	[77]
MSCs	No application	Not clear	Graphene oxide doped with Fe_3_O_4_	Promoting osteogenic differentiation in presence of BMP2	[78]
Human BMSCs	Rabbit model of femoral bone defects	15 mT	Polydopamine doped with Fe_3_O_4_	Enhancing cell proliferation and osteogenic differentiation in vitro and new bone formation in vivo	[79]
Pre-osteoblasts MC3T3-E1	No application	200 mT	CoFe_2_O_4_/P(VDF-TrFE) nanocomposite coatings	Enhancing cell adhesion, proliferation, and differentiation	[80]
Pre-osteoblasts MC3T3-E1	No application	200 mT	ZnFe_2_O_4_ coatings	Promoting early proliferation (3 days) and osteogenic differentiation	[81]
Rat BMSCs	Rat model of femoral bone defects	50 mT	SPIONs were encapsulated into PLGA microspheres	Promoting osteogenic differentiation in vitro and repairing bone defects in vivo	[82]
Rat BMSCs	No application	15 mT	Graphene oxide doped with Fe_3_O_4_	Promoting osteogenesis in BMSCs	[83]
Pre-osteoblasts MC3T3-E1 and pre-osteoclasts 4B12	No application	200 mT	PMMA covered Co_0_._5_Mn_0_._5_Fe_2_O_4_	Promoting osteoblastic differentiation and modulating osteoclastogenesis	[84]
Pre-osteoblasts MC3T3-E1 and pre-osteoclasts 4B12	No application	200 mT	Ca_5_(PO_4_)_3_OH/Fe_3_O_4_ functionalized with microRNAs	Activating osteogenesis and inhibiting osteoclastic differentiation	[85]

PLA: poly lactide acid; BSA: bovine serum albumin; TPU: thermoplastic polyurethane; PCL: polycaprolactone; PLGA: poly(lactide-co-glycolide); MSCs: mesenchymal stem cells; BMSCs: bone marrow mesenchymal stem cell; hDPSCs: human dental pulp stem cells; ADSCs: adipose-derived mesenchymal stem cells; PMMA: polymethacrylate.

## Data Availability

Not applicable.

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
