# Peer review of "Iron Oxide Nanoparticles Combined with Static Magnetic Fields in Bone Remodeling"

_cells, 2022, doi:10.3390/cells11203298_

Round 1
Reviewer 1 Report (Previous Reviewer 1)
The authors have done a lot of work to systematize the latest data on the role of IONPs and SMFs in bone remodeling. Self-citation [43] to the review DOI: 10.1016/j.arr.2022.101717 should be excluded from the References. The link contains well-known information: "Iron is an essential element involved in multiple life activities of the human body, including bone metabolism [43]."
Response: Dear reviewer, we appreciate you very much for your comments on our manuscript. For your kind comments, we respond as follows,
Although “Iron is an essential element involved in multiple life activities of the human body” is a well-known information, “iron and bone metabolism” is not well-known. Thus we think that references [43] should be retained.
Reviewer 2 Report (Previous Reviewer 3)
The authors addressed all my concerns
Response: Dear reviewer, we appreciate you very much for your warm work on our manuscript.
This manuscript is a resubmission of an earlier submission. The following is a list of the peer review reports and author responses from that submission.
Round 1
Reviewer 1 Report
This is an interesting review and after revision it will be of interest to a wide range of readers.
1. Section 5. Mechanism of static magnetic field enhanced biological effects of iron oxide nanoparticles requires a scheme.
2. It is necessary to discuss such a phenomenon as ferroptosis
3. A chapter should be added - methods for obtaining nanoparticles.
Author Response
Dear reviewer, we appreciate you very much for your positive and constructive comments on our manuscript. For your kind comments, we respond step by step as follows,
Point 1: Section 5. Mechanism of static magnetic field enhanced biological effects of iron oxide nanoparticles requires a scheme.
Response 1: We have added a scheme in the section 5, that is Figure 2.
Point 2: It is necessary to discuss such a phenomenon as ferroptosis
Response 2: We have a paragraph to discuss ferroptosis in the section 6.
Point 3: A chapter should be added - methods for obtaining nanoparticles.
Response 3: The methods for obtaining nanoparticles have been reported detailedly in abundant review articles, such as “Li S, Wei C, Lv Y. Preparation and Application of Magnetic Responsive Materials in Bone Tissue Engineering. Curr Stem Cell Res Ther, 2020, 15(5): 428-440.”; “Fan D, Wang Q, Zhu T et al. Recent Advances of Magnetic Nanomaterials in Bone Tissue Repair. Front Chem, 2020, 8: 745.”; “Dadfar SM, Roemhild K, Drude NI, von Stillfried S, Knüchel R, Kiessling F, Lammers T. Iron oxide nanoparticles: Diagnostic, therapeutic and theranostic applications. Adv Drug Deliv Rev, 2019, 138: 302-325.”, etc. Therefore, we think that the description of the preparation method of nanoparticles here is somewhat repetitive with the previous articles.
Reviewer 2 Report
General comments
The article is a review article that revise the effects of iron oxide nanoparticles and the intensities of static magnetic forces in bone remodelling, thus analyzing effects of particles in osteoblasts and osteoclasts biology and function. The authors collected enough data, citing an appropriate number of articles, including the latest advances in the field and possible applications of this technology.
Unfortunately, the article has numerous grammar errors that difficult the understanding of the work. Some paragraphs lack of a rationale flow, and most of them are written just as successive highlights of results of specific articles, without conclusions, comments or meaning that would help the reader to follow the scientific story that the authors aim to build.
I encourage to go through a thorough prof reading, and rearrangement of sentences within paragraphs. Below I include comments on some sections and provide examples of how a review like this could be presented.
Specific comments:
Line 29, describes properties of iron oxides NPs, citations are needed.
Line 53-54, grammar problem
Line 55-57, grammar problem.
Line 58-59 osteoblasts and osteoclasts are plural
Line 67, grammar again: osteogenesis of osteoblasts? Meaning that a machine is osteogening osteoblasts?
Line 78, we refer to MG-63 as a cell line.
Line 79, “cell mineralization” the referred article does not show mineralization of cells, which would be catastrophic. Again it’s a grammar error. It can be used “cell-induced mineralization”, “biomineralization”, etc.
Line 87-91 sentence too long. Hard to read. Break it up.
Paragraph 99-115 Sounds like a bunch of citations, without interpretation, conclusion, suggestion or meaning that help the flow or the reader to enjoy a scientific story. Grammar errors in each sentence does not allow the flow. Needs major work. Bellow you can find my comment on each sentence. Sentences are color coded with the comments at the end of this review to help the authors work on this article prior to resubmission.
In line with in vitro studies, numerous in vivo studies have shown that scaffolds complexed with IONPs can promote bone formation. Hu et al.[20] implanted the IONPs loaded gelatin sponge into the incisor sockets of rats, found that the formation of new bone was significantly promoted after 4 weeks. Liao et al.[21] showed that IONPs can promote the differentiation of human precartilaginous stem cells (hPCSCs) into osteoblasts in vitro, and hPCSCs combined with IONPs can significantly accelerate bone formation in bone defects and promote bone regeneration in vivo. Xia et al.[22] incorporated γFe2O3 and αFe2O3 nanoparticles into calcium phosphate cement (CPC) scaffolds, and seeded human dental pulp stem cells (hDPSCs) on IONP-CPC scaffolds and found that the osteogenic differentiation and bone mineral deposition of hDPSCs was significantly enhanced. Similarly, Fe3O4-incorporated IONP-CPC scaffolds also enhanced the osteogenic differentiation of hDPSCs, and promoted mandibular bone defect repair in rats[23]. Mechanistically, IONP-CPC scaffolds promote osteogenic differentiation of hDPSCs by activating Wnt/β-catenin signaling[24]. Moreover, Xia et al.[25] investigated the effects of IONPs as a liquid or powder on hDPSCs using IONP-CPC scaffolds. They showed both forms of IONPs significantly enhanced osteogenesis in hDPSCs, and the IONPs incorporated in the liquid form had a better osteogenesis effect than the IONPs in the powder form.
Initial statement. I understand here that you will refer to in vivo studies to confirm what has been propose in vitro.
That’s it? no comment, no interpretation, no other results, no details. The particles in this work were super paramagnetic NPs coated with PSC, without application of external magnetic force. the results were observed by MRI and histology. There were control groups, etc. THey increased the osteogenic phenotype of cells quantified by OC secretion. Really poor presentation and analysis of this work to support the initial statement.
Here is an idea of how to present this paper in the paragraph “Hu et al. Implanted Superparamagnetic IONPs by use of loaded gelatinous sponges in rat’s incisor sockets. By MRI and histological observation, they found increased bone formation after 4 weeks, suggesting that these NPs increase osteoblast activity and endothelial function in vivo…”
The in vivo part does not specify that study was made in rabbit. Could be critical since it is correctly mentioned the the invitro part of the study was done using human cells, then the in-vivo part was done in a human?
Again, what type of particles?
Again, poor grammar, hard to follow here is an idea of how to present this sentence “…generated a scaffold by incorporating γFe2O3 and αFe2O3 nanoparticles into calcium phosphate cement (CPC). They found that human dental pulp cells seeded in this scaffold showed increased osteogenic differentiation, ALP secretion and mineral matrix synthesis compared to those seeded in scaffolds without IONPs, demonstrating the effects of these particles on…”
This is a well suited and well cited article to confirm previous description.
Sentence out of context in this paragraph. The objective of the paragraph was to present in-vivo work that would confirm in-vitro results. We can talk about mechanisms, which are partially known in a different section. Furthermore, the sentence is not followed by more mechanism analysis but by the physical form in which NPs are presented, liquid or solid. which confirms that there is no clarity about the objective of the whole paragraph.
“Moreover” seems to be a continuing sentence that will further justify the previous one, but instead this one refers to the physical properties of the particles, not connected at all with the mechanistic.
Author Response
Dear reviewer, we appreciate you very much for your positive and constructive comments on our manuscript. For your kind comments, we respond step by step as follows,
Point 1: Line 29, describes properties of iron oxides NPs, citations are needed.
Response 1: We have added a citation.
Point 2: Line 53-54, grammar problem
Response 2: We have corrected and marked word with red color in the manuscript.
Point 3: Line 55-57, grammar problem.
Response 3: We have corrected and marked word with red color in the manuscript.
Point 4: Line 58-59 osteoblasts and osteoclasts are plural
Response 4: We have corrected.
Point 5: Line 67, grammar again: osteogenesis of osteoblasts? Meaning that a machine is osteogening osteoblasts?
Response 5: We have corrected “osteogenesis of osteoblasts” to “osteogenic differentiation”.
Point 6: Line 78, we refer to MG-63 as a cell line.
Response 6: We have corrected “osteoblast MG-63” to “MG-63 cells, an osteoblast cell line, ….”.
Point 7: “cell mineralization” the referred article does not show mineralization of cells, which would be catastrophic. Again it’s a grammar error. It can be used “cell-induced mineralization”, “biomineralization”, etc.
Response 7: We have deleted the word “cell mineralization” in Ref. 13 and gone through a thorough prof reading to correct “cell mineralization” to “biomineralization”.
Point 8: Line 87-91 sentence too long. Hard to read. Break it up.
Response 8: We have broken up the long sentence to two short sentences, and marked them with red color.
Point 9: Paragraph 99-115 Sounds like a bunch of citations, without interpretation, conclusion, suggestion or meaning that help the flow or the reader to enjoy a scientific story. Grammar errors in each sentence does not allow the flow. Needs major work. Bellow you can find my comment on each sentence. Sentences are color coded with the comments at the end of this review to help the authors work on this article prior to resubmission.
That’s it? no comment, no interpretation, no other results, no details. The particles in this work were super paramagnetic NPs coated with PSC, without application of external magnetic force. the results were observed by MRI and histology. There were control groups, etc. THey increased the osteogenic phenotype of cells quantified by OC secretion. Really poor presentation and analysis of this work to support the initial statement.
Here is an idea of how to present this paper in the paragraph “Hu et al. Implanted Superparamagnetic IONPs by use of loaded gelatinous sponges in rat’s incisor sockets. By MRI and histological observation, they found increased bone formation after 4 weeks, suggesting that these NPs increase osteoblast activity and endothelial function in vivo…”
The in vivo part does not specify that study was made in rabbit. Could be critical since it is correctly mentioned the the invitro part of the study was done using human cells, then the in-vivo part was done in a human?
Again, what type of particles?
Again, poor grammar, hard to follow here is an idea of how to present this sentence “…generated a scaffold by incorporating γFe2O3 and αFe2O3 nanoparticles into calcium phosphate cement (CPC). They found that human dental pulp cells seeded in this scaffold showed increased osteogenic differentiation, ALP secretion and mineral matrix synthesis compared to those seeded in scaffolds without IONPs, demonstrating the effects of these particles on…”
This is a well suited and well cited article to confirm previous description.
Sentence out of context in this paragraph. The objective of the paragraph was to present in-vivo work that would confirm in-vitro results. We can talk about mechanisms, which are partially known in a different section. Furthermore, the sentence is not followed by more mechanism analysis but by the physical form in which NPs are presented, liquid or solid. which confirms that there is no clarity about the objective of the whole paragraph.
“Moreover” seems to be a continuing sentence that will further justify the previous one, but instead this one refers to the physical properties of the particles, not connected at all with the mechanistic.
Response 9: We have checked all sentences in Paragraph 99-115 and corrected them in terms of your comment on each sentence. Corrected sentences are marked with red color. In addition, we added several missing in vivo studies in the manuscript.
Reviewer 3 Report
The paper is interesting and well written, only few concerns should be addressed:
in the introduction ashot paragraph should be added on osteoblastogenesis and osteoclastogenesis, highlighting the main pathway cited in the text.
Author Response
Dear reviewer, we appreciate you very much for your positive and constructive comments on our manuscript. For your kind comments, we respond step by step as follows,
Point 1: in the introduction ashot paragraph should be added on osteoblastogenesis and osteoclastogenesis, highlighting the main pathway cited in the text.".
Response 1: We have added descriptions and explanations of the main pathways involved in osteoblastogenesis and osteoclastogenesis, not in the introduction, but in the paragraph in which the pathway is cited.
Round 2
Reviewer 2 Report
General comments:
The authors made substantial changes in the manuscript that significantly improved its quality. However, they didn’t revise the whole article (as recommended by this reviewer) and there are sections that still needs work on the structure and grammar before publication. In my previous comments I wrote “Unfortunately, the article has numerous grammar errors that difficult the understanding of the work. Some paragraphs lack of a rationale flow, and most of them are written just as successive highlights of results of specific articles, without conclusions, comments or meaning that would help the reader to follow the scientific story that the authors aim to build. I encourage to go through a thorough prof reading, and rearrangement of sentences within paragraphs. Below I include comments on some sections and provide examples of how a review like this could be presented.” Although my specific comments and examples limited to the first three paragraphs, they are extensive to the whole article that have the same issues all along the text. The authors didn’t edit the whole manuscript that have still the same problems in large sections of it. More examples of this are shown below.
When the grammar or flow of an original article are the main problem, the reviewer can be flexible if the science, logic and experiments and figures are well designed and organized, but in a literature review the organization of ideas and flow are the main part. It is veer unfortunate that the message that I provided previously was not clear enough.
Specific comments:
Paragraph starting at 232:
What is the message? Conclusion? Any authors’ thoughts other than providing a list of results from published articles?
258- What do osteocytes do in a section about osteoclasts? Even in the wrong lineage! Really bad signal to a reviewer.
266 -269 grammar problems
Paragraph starting at 275
Again, a list of results from different articles. What is the flow here? This can be perfectly converted into a table. There is no intellectual activity here, just providing a list of results. What can be concluded form the articles that led to the next step?
I will stop here. Does not worth my time to follow down. Sorry to be rude, this is the second time I made the same comment
Author Response
Dear reviewer, we appreciate you very much for your positive and constructive comments on our manuscript. For your kind comments, we respond step by step as follows,
Point 1: Paragraph starting at 232:
What is the message? Conclusion? Any authors’ thoughts other than providing a list of results from published articles?
Response 1: We rearranged the order of all sentences in Paragraph 232-263 and added some conclusions, comments and other sentences to make the whole section logical. Corrected sentences are marked with red color.
Point 2: 258- What do osteocytes do in a section about osteoclasts? Even in the wrong lineage! Really bad signal to a reviewer.
Response 2: Since osteocytes are derived from mature osteoblasts, thus we moved the paragraph about osteocytes to the section on osteoblasts.
Point 3: 266 -269 grammar problems.
Response 3: We have corrected and marked word with red color in the manuscript.
Point 4: Paragraph starting at 275
Again, a list of results from different articles. What is the flow here? This can be perfectly converted into a table. There is no intellectual activity here, just providing a list of results. What can be concluded form the articles that led to the next step?
Response 4: We rearranged the order of all sentences in Paragraph 275-340 and added some conclusions, comments and other sentences to make the whole section logical. Corrected sentences are marked with red color.
In addition, the manuscript was checked throughout the article and some spelling and grammatical errors were corrected.
Round 3
Reviewer 2 Report
The authors made substantial changes to the manuscript, which is now in a better suited form, more clear and with a better flow.